# Molecular Features and Diagnostic Challenges in Alpha/Beta T-Cell Large Granular Lymphocyte Leukemia

**DOI:** 10.3390/ijms232113392

**Published:** 2022-11-02

**Authors:** Francesco Gaudio, Pierluigi Masciopinto, Emilio Bellitti, Pellegrino Musto, Elena Arcuti, Olga Battisti, Gerardo Cazzato, Alessandra Solombrino, Filomena Emanuela Laddaga, Giorgina Specchia, Eugenio Maiorano, Giuseppe Ingravallo

**Affiliations:** 1Hematology Section, Department of Emergency and Transplantation, University of Bari Medical School, 70124 Bari, Italy; 2Section of Pathology, Department of Emergency and Organ Transplantation (DETO), University of Bari Aldo Moro, Piazza G. Cesare, 11, 70124 Bari, Italy; 3Clinical Pathology Unit, AOUC Policlinico, Piazza G. Cesare, 11, 70124 Bari, Italy; 4School of Medicine, University of Bari “Aldo Moro”, Piazza G. Cesare, 11, 70124 Bari, Italy

**Keywords:** leukemia, molecular features, clonal lymphocyte expansion, T-cell-derived LGL

## Abstract

Large granular lymphocyte leukemia is a rare chronic lymphoproliferative disease of cytotoxic lymphocytes. The diagnosis, according to the WHO, is based on a persistent (>6 months) increase in the number of LGL cells in the peripheral blood without an identifiable cause. A further distinction is made between T-LGL and NK-LGL leukemia. The molecular sign of LGL leukemia is the mutation of STAT3 and other genes associated with the JAK/STAT pathway. The most common clinical features are neutropenia, anemia, and thrombocytopenia, and it is often associated with various autoimmune conditions. It usually has an indolent course. Due to the rarity of the disease, no specific treatment has yet been identified. Immunosuppressive therapy is used and may allow for disease control and long-term survival, but not eradication of the leukemic clone. Here, we discuss the clinical presentation, diagnostic challenges, pathophysiology, and different treatment options available for alpha/beta T-LGL leukemia, which is the most common disease (85%), in order to better understand and manage this often misunderstood disease.

## 1. Introduction

Large granular lymphocytes (LGL) are a morphologically distinct subpopulation of peripheral blood mononuclear cells, which normally account for 10–15% of peripheral blood mononuclear cells [1,2,3,4].

T-LGL leukemia exhibits a CD3+CD8+CD57+TCRαβ+ phenotype in most cases, signifying a constitutively activated T-cell phenotype. Molecular analysis of the clonal repertoire of the T-cell receptor (TCR) in LGL leukemia is important for identifying clonal expansion and monitoring clinical response [1,2].

The etiology of T-LGL leukemia remains unknown. It has been suggested that an initial antigen-driven T-cell expansion, followed by the occurrence of oncogenic events (i.e., somatic mutations), leads to the persistence of abnormal T-cell clones [1,5,6].

Activation of alpha/beta T-LGL strictly depends on their recognition of the specific MHC peptide complex, and they act mainly on the adaptive immune system. Instead, gamma/delta T-LGL have a limited TCR variety and possess intact antigenic responsiveness. This feature places them between innate and adaptive immune responses. Unlike the immediate cytotoxic activity of NK cells, both alpha/beta and gamma/delta T-LGL require an activation step to become cytotoxic. After activation, LGLs can hit the target via the granule-mediated or receptor-mediated pathways [7,8,9,10].

Clonal expansion is usually a hallmark of malignancy. However, clonal expansion of lymphocytes may be a normal process that develops after meeting the antigen. Therefore, it is necessary to differentiate a transient immune response from persistent malignancy [11].

Apart from rare aggressive cases, most patients with LGL leukemia have an indolent clinical course, and about one-third of them are asymptomatic at diagnosis [12].

Recent studies have identified recurrent mutations affecting the JAK/STAT pathway in many malignant T and NK cell tumors, highlighting the overlapping biology in many of these neoplasms. STAT3 mutations are evident in both large granular lymphocytic leukemias (T and NK cells). STAT5B mutations are rarer and are associated with a more aggressive disease course [13,14,15].

Here, we summarize the clinical and molecular features and the diagnostic challenges of LGL leukemia, examining the associated clinical conditions and causal relationships.

## 2. Cytology

A persistently increased number of circulating LGLs for more than six months must be identified in the peripheral blood for diagnosis. A lower number may be compatible with the diagnosis if the clonality of these cells is demonstrated and if it is associated with particular clinical or hematological conditions.

Although leukemic LGLs are easily identifiable in peripheral blood due to their characteristic morphology, they are not cytologically distinguishable from normal reactive cytotoxic lymphocytes. LGLs are large cells (15–18 m) with a kidney-shaped or round nucleus and abundant cytoplasm containing azurophilic granules [3].

### 2.1. Immunophenotype

T-LGL leukemias have a constitutive mature post-thymic phenotype. The most frequent phenotype is of constitutively activated T cells (CD3+, TCR αβ+, CD4−, CD5dim, CD8+, CD16+, CD27−, CD28−, CD45R0−, CD45RA+, and CD57+), which are aggressive and generally associated with Stat5b mutations [3,16].

CD41 positive LGL leukemia with or without CD8 co-expression almost never features cytopenias, splenomegaly, or autoimmune phenomena and has been associated with Stat5b mutations. T-LGL leukemics show a terminal effector memory phenotype with CD45RA expression and CD62L deficiency.

### 2.2. Clonality

Evidence of T-LGL clonality is routinely assessed using TCR g-polymerase chain reaction assays. TCR gene repertoire analysis can also be ascertained using flow cytometry, which serves as presumptive evidence for clonality. The current Vb mAbs panel covers 75% of the Vb spectrum, showing a high correlation with TCRg-polymerase [3].

### 2.3. WHO Classification

LGL leukemia is classified by the WHO under mature T and NK cell neoplasms, subdivided into three categories: T-LGL leukemia, NK-LGL leukemia, and aggressive NK cell leukemia (ANKL) [2].

According to the diagnostic criteria of the World Health Organization (WHO), in order to diagnose LGL leukemia, LGL must be increased for a period of more than six months.

The clinical and biological features of T-LGL leukemia and CLPD-NK are similar, while ANKL is characterized by multiorgan infiltration by LGL and a rapidly fatal course.

Molecular pathways involved in the pathogenesis of LGL leukemia.

Figure 1 shows the molecular pathways involved in the pathogenesis of LGL leukemia.

*JAK-STAT3 signaling pathway:* The JAK receptor has a tyrosine kinase activity that, by binding to its ligand (IL-15), undergoes autophosphorylation, initiating the binding of STAT protein and phosphorylation by JAK, and then undergoes dimerization. STAT dimers enter the nucleus and regulate gene expression, particularly genes related to cell survival. The mutation of this pathway has been demonstrated to be a key feature in the pathogenesis of LGL leukemia, promoting the survival of LGL [17]. Mutations in the activated domain of the STAT3 gene have been observed in 30–40% of patients with LGL leukemia [17,18].

The proinflammatory cytokine IL-6 induces, while the cytokine-3 signaling suppressor (SOCS3) inhibits, the Jak-Stat3 pathway in patients with LGL leukemia [19]. Furthermore, IL-6 levels are increased, while messenger RNA and SOCS3 protein levels are significantly reduced in LGL leukemia. The STAT3 mutant can predict poor prognosis and contribute to the development of associated disorders, such as aplastic anemia and myelodysplastic syndrome [18,19].

*FAS/FAS-L mediated pathways:* The Fas and Fas-L signaling pathways normally induce apoptosis and play an important role in regulating the immune system. They work through the activation of caspase-dependent apoptosis. This is the main mechanism by which LGLs induce cell death of foreign cells. Normal LGLs, once activated due to infection, resist apoptosis, and when the infection clears, these activated LGLs are eliminated through activation-induced cell death (AICD) of the FAS-FAS-L pathway. Unlike regular LGLs, leukemic LGLs are resistant to AICD. A cleaved FAS product known as soluble FAS can block AICD by interfering with the normal binding of Fas-L to its receptor. Fas-L is responsible for hematopoietic suppression through the apoptosis of neutrophil precursors in the bone marrow. This is responsible for the development of neutropenia, the characteristic clinical presentation in patients with LGL leukemia [20,21].

*RAS-RAF-1-MEK1-ERK pathway:* The RAS pathway is involved in the survival, proliferation, senescence, and differentiation of normal cells. Activation of RAS results in the activation of RAF by phosphorylation, followed by the activation of MEK and ERK in sequence. The mutation of RAS results in abnormal cell signals, the deregulation of gene expression, and oncogenesis [22]. Hyperactivation of the Ras pathway plays an important role in LGL survival signaling. Co-activation of Ras and ERK was found in LGL leukemia [23]. Blocking ERK or Ras activity can restore sensitivity to Fas in leukemic LGL [24]. This can be used as a likely therapeutic target.

*PI3K/AKT pathway:* PI3K is a downstream member of the growth factor receptor with tyrosine kinase activity, which acts through the phosphorylation of AKT and thus its activation. Activated AKT participates in cell survival, proliferation, and growth [25]. Overexpression of the PI3K-AKT signaling pathway was found in T-LGL cells and is associated with inhibition of apoptosis [26].

*NF-KB pathway:* NF-KB is a transcription factor involved in the survival of immune cells. It remains silent by binding to its inhibitors, known as NF-KB inhibitors, in resting conditions. After activation, NF-KB is translocated to the nucleus, where it regulates the expression of various genes. NF-KB is activated in LGL leukemia and acts by enhancing the anti-apoptotic function of Bcl-2 proteins. It also acts through the PI3K-AKT pathway to prevent apoptosis through Mcl-1, independently of STAT3 [27].

*Sphingolipid rheostat pathway:* Sphingolipids, sphingosine-1-phosphate (S1P), and ceramide are interconvertible metabolites that exist in a balanced state in the blood. The increase in S1P is responsible for the increase in cell survival, while increased levels of its metabolite ceramide lead to apoptosis. In patients with LGL leukemia, there is an increase in S1P [28].

## 3. Case Presentation

LGL leukemias are commonly diagnosed in elderly patients, with a median age at diagnosis of over 60 years. At diagnosis, only 15% of patients are under the age of 50. The incidence is the same in men and women, but in women the diagnosis is more often made at a young age [29].

The most frequent clinical presentation is characterized by recurrent infections related to chronic neutropenia, while in one-third of patients the disease is asymptomatic. One-quarter of patients have splenomegaly, rarely hepatomegaly or lymphadenopathy. Fatigue and B symptoms are present in 20–30% of cases [1,30].

## 4. Diagnostic Challenges in LGL Leukemia

LGL counts below 2 × 10^9^/L that meet the diagnosis are still consistent. Hence, this diagnosis can only be made once clone persistence has been demonstrated (see Figure 2, Figure 3 and Figure 4).

Cases of LGL leukemia with more unusual immunophenotypic profiles, such as the gamma/delta TCR, can lead to other differential diagnoses, such as gamma/delta hepatosplenic T-cell lymphoma [31]. However, the two diseases can be distinguished by combining clinical features, such as generalized symptoms, rapid onset of symptoms, the presence of hepatosplenomegaly, and the sinusoidal bone marrow expansion of lymphoma cells. The identification of a T-cell clone is not synonymous with a neoplasm. T-cell clones can be detected due to reactive causes, infection, and senescence and can be persistent [32].

There is a spectrum of both infectious and autoimmune diseases that can be associated with the polyclonal expansion of T lymphocytes, monoclonal expansion, and neoplastic proliferation. Furthermore, the chronic activation of T lymphocytes by viruses, such as HTLV or EBV, has been linked to the pathogenesis of lymphoproliferative disorders, such as LGL leukemia. This boundary and how we define these clones can be complex and can change over time [10]. Additionally, with improvements in diagnostics, the number of individuals identified with persistent T-cell clones with normal or even low lymphocyte counts will increase. Interestingly, there is an association between clonal hematopoiesis (CHIP), myelodysplastic syndrome (MDS), and clonal T-cell diseases, which share many of the same recurrent mutations seen predominantly in epigenetic regulators, such as DNMT3A and TET2, suggesting that they may be early mutations in common progenitors. There are numerous reports of MDS coexisting with LGL leukemia or angioimmunoblastic T-cell lymphoma [33]. The significance of T-cell clones in the context of cytopenias, and thus how to manage them, is unclear [5,9]. It should also be noted that LGL proliferations may be observed with other hematological and non-hematological conditions [7,8]. The presence of mutations in STAT3 and STAT5b does not immediately define a diagnosis of LGL leukemia, as these mutations are not specific for this disease [10]. If patients do not have sufficient evidence for a diagnosis of T-LGL, they can over time acquire secondary events that drive progression, ultimately turning into malignant tumors.

## 5. Disorders Related to Indolent LGL Leukemia

### 5.1. Hematological Disorders

The hematological disorders associated with LGL leukemia, and their frequency, are shown in Table 1. Less than half of these patients have lymphocyte counts ranging from 4 × 10^9^/L to 10 × 10^9^/L. In 7–36% of cases, the LGL count may be lower (from 0.5 × 10^9^/L to 1 × 10^9^/L).

Neutropenia is present in 50% of patients, and severe neutropenia is present in about 20% [34]. Other hematological problems that can occur in LGL leukemia are transfusion-dependent anemia, which affects between 6% and 22% of patients, and pure red cell aplasia, which occurs in 8–19% of cases. Hemolytic anemia may be present with either a positive or negative Coombs tests due to a direct inhibitory effect of leukemic LGL on erythropoiesis. Thrombocytopenia is described in less than 20% of cases and has been associated with the clonal suppression of megakaryopenia [35,36,37].

**Table 1 ijms-23-13392-t001:** Hematological disorders in LGL leukemia. Data from the two largest retrospective series [35,36].

Splenomegaly	(24%)
Median LGL × 10^9^/L (range)	1.7 (0.8–3.3)
ANC < 1.5 × 10^9^/L	(46–59%)
ANC < 0.5 × 10^9^/L	(17–24%)
Anemia Hb < 11 g/dL	(24–40%)
Hb < 8 g/dL	(6–22%)
Thrombocytopenia	(17–30%)
MGUS	(10–20%)

Abbreviations: LGL, large granular lymphocyte; ANC, absolute neutrophil count; HB: hemoglobin; MGUS: monoclonal gammopathy of undetermined significance.

The positivity of rheumatoid factor (40–60%), antinuclear antibodies (40%), anti-neutrophil antibodies (20–60%), or direct Coombs underlines the immune context of this lymphoproliferative disorder. Hypogammaglobulinemia is present in 5–10% of patients. Defects in the regulation of immunoglobulin secretion in LGL leukemia could partly explain the development of autoantibodies and clonal B-cell tumors observed in this disease (monoclonal gammopathy of undetermined significance is associated in 10–20% of cases).

In myelodysplastic syndrome, abnormal bone marrow progenitors have been shown to initiate clonal expansion of LGL which causes immunopathological damage to circulating blood cells and leads to a deficient proliferation state, such as neutropenia, thrombocytopenia, aplastic anemia, and red blood cells pure aplasia [38].

Bone marrow analysis can be difficult because the leukemic population is often small and immunopathological damage can create a hypercellular or hypocellular marrow (Figure 2 and Figure 3), depending on the host response [1,39].

### 5.2. Autoimmune Disorders

LGL leukemia is associated with a broad spectrum of autoimmune diseases (15–40%), often involving connective tissue (Table 2) [1,36,37,40,41,42,43,44].

These disorders can be subdivided into the following: arthropathies, vascular and skin disorders, neuropathies, muscle and glandular disorders, and other connective tissue disorders. Rheumatoid arthritis (RA) is present in about 15% of cases. Systemic lupus erythematosus, Sjögren’s syndrome, autoimmune thyroid disorders, coagulopathy, and inclusion body myositis have occasionally been reported. Vasculitis with cryoglobulinemia has also been reported. Furthermore, cases of pulmonary arterial hypertension, considered as a vascular disease with endothelial dysfunction, have been reported to be associated with LGL leukemia [11,29,30,41].

Infections secondary to chronic neutropenia mainly involve the skin, oropharynx, and perirectal area and affect 15% to 39% of patients. Serious septic complications, which are the leading cause of related death, can occur, affecting approximately 5% to 10% of patients [1,39].

## 6. Hematological Diseases Associated with LGL

LGL leukemia has been associated with other hematological diseases, such as myelodysplastic syndromes (MDS), aplastic anemia (AA), pure red cell aplasia (PRCA), paroxysmal nocturnal hemoglobinuria (PNH), myeloproliferative neoplasm, B-cell neoplasm (Table 3), with a frequency of 3–10% [38,45,46,47,48,49,50].

In a patient with unexplained cytopenias, the potential presence of MDS and/or LGL leukemia should be evaluated. However, the final diagnosis can be a diagnostic challenge, especially in cases without certain characteristics of MDS (e.g., typical cytogenetics, excess blasts, ring sideroblasts), or in the case of LGL, leukemia with lymphocytosis [19,51].

## 7. Transplant of Solid Organs and Association with LGL

In solid organ transplantation, the allograft-driven immune response, immunosuppressive treatments, and concomitant infections create a favorable situation for the expansion of clonal lymphocytes. In 5% of cases, this proliferation can involve the T/NK line [52].

A diagnosis of LGL leukemia in a transplant recipient does not prognostically appear to have a worse clinical outcome, and the proportion of patients requiring specific therapy is similar to that described in non-transplanted LGL leukemia patients (4060%), although the majority of transplant recipients are usually on immunosuppressive treatment.

## 8. Therapy

There is no standard treatment for patients with LGL leukemia. Indications for treatment include severe neutropenia (absolute neutrophil count <0.5 × 10^9^/L), moderate neutropenia (ANC > 0.5 × 10^9^/L) associated with recurrent infections, symptomatic or transfusion-dependent anemia, and associated autoimmune conditions requiring therapy. However, it is clear that immunosuppressive therapy remains the foundation of treatment (Figure 5), including the individual agents methotrexate (MTX), oral cyclophosphamide, and cyclosporine A (CyA). Nevertheless, immunosuppressive agents, such as MTX, cyclophosphamide, and CyA, are limited in their ability to eradicate the LGL clone and induce long-lasting remission. Due to the lack of prospective comparative clinical trials, it is not possible to decide which of these three agents is best proposed as first-line therapy [30,53,54].

MTX is the most commonly applied therapy in neutropenic patients, while cyclophosphamide is preferred in patients with anemia and PRCA. However, treatment for more than 12 months with cyclophosphamide should be avoided due to the mutagenic potential. Several clinical studies have evaluated the effectiveness of new therapeutic combinations, such as thalidomide, prednisone, and methotrexate (TPM). The results of a prospective analysis showed 18/20 (90%) responses with 80% complete remissions.

For patients unresponsive to a first line, a switch from MTX to cyclophosphamide may be useful; alternatively, treatment with purine analogs showed promise, with an overall response of 79%. Alemtuzumab, a humanized anti-CD52 monoclonal antibody, has been studied in refractory patients, showing an overall response close to 60% [4,55].

Novel therapeutic approaches in recent studies have evaluated Hu-Mikβ 1, a humanized mAb directed at the shared IL-2/IL-15Rβ (CD122) subunit, without valid results, and bortezomib for constitutional activation of Nf-Kb [56].

Inhibition of Jak/Stat3 could be a good therapeutic option in patients with LGL leukemia. Tofacitinib citrate (CP690550), a specific inhibitor of Jak3, has been tested in patients with refractory LGL leukemia and rheumatoid arthritis, and an improvement in neutropenia has been observed [57].

OPB-11, another Stat inhibitor, has demonstrated its efficacy against leukemia cells with STAT-addictive oncokinase [58].

Finally, the new BNZ-1 multi-cytokine inhibitor has shown promising results in patients with LGL leukemia. It selectively inhibits IL-2 and IL-15 and, to a lesser extent IL-9, without affecting IL-4, IL-7, or IL-21 [59].

## 9. Conclusions

LGL leukemia is characterized by the abnormal clonal expansion of mature LGLs that remain competent over the long term. Tissue invasion is a common feature of leukemic LGL, leading to a broad spectrum of immune-based hematological and non-hematological diseases. This review outlined clinical presentations in which the presence of LGL leukemia should be suspected. We also discussed the mechanisms behind these associations. The association between malignancy and autoimmunity suggests the possibility of a common pathogenesis in LGL leukemia and many hematological disorders, as well as in autoimmune diseases. The application of new molecular techniques, together with immunological markers, has greatly aided the understanding of their pathogenesis and provided the basis for the development of new treatments.

## Figures and Tables

**Figure 1 ijms-23-13392-f001:**
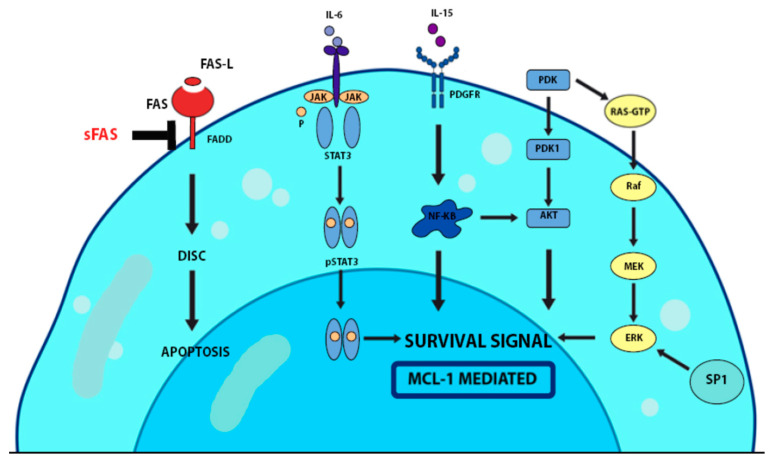
Molecular pathways involved in LGL leukemia.

**Figure 2 ijms-23-13392-f002:**
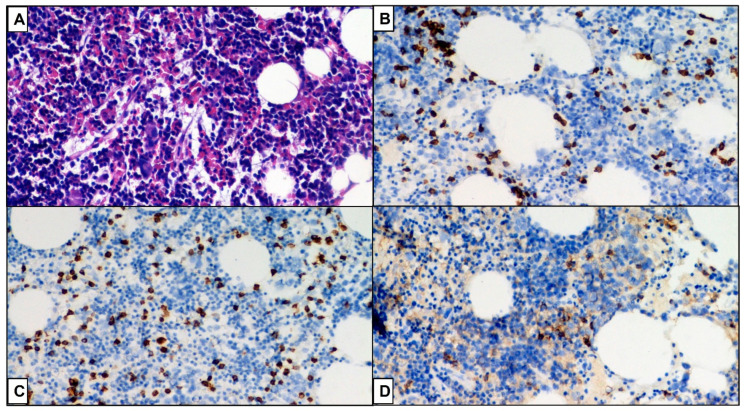
Hypercellular bone marrow with erythroid hyperplasia (>50% of total BM cells). (**A**) Hematoxylin-eosin, original magnification 100×. (**B**) Immunoreactivity for CD3, immunohistochemistry, 20×. (**C**) Immunoreactivity for CD8, immunohistochemistry, 100×. (**D**) Immunoreactivity for CD4, immunohistochemistry, 100×.

**Figure 3 ijms-23-13392-f003:**
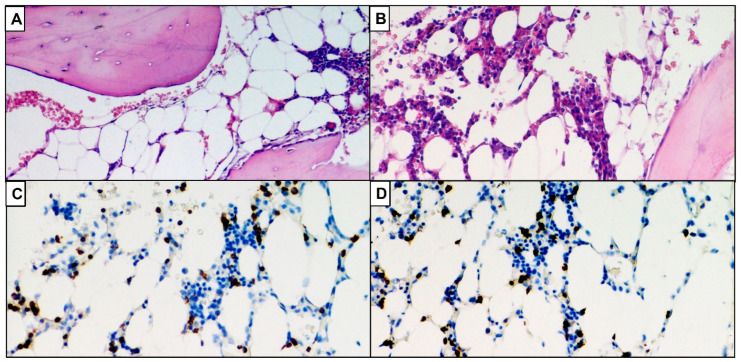
Remarkably hypocellular bone marrow. (**A**,**B**) Hematoxylin-eosin, original magnification 100×. (**C**) Immunoreactivity for CD3, immunohistochemistry, 100×. (**D**). Immunoreactivity for CD8, immunohistochemistry, 100×.

**Figure 4 ijms-23-13392-f004:**
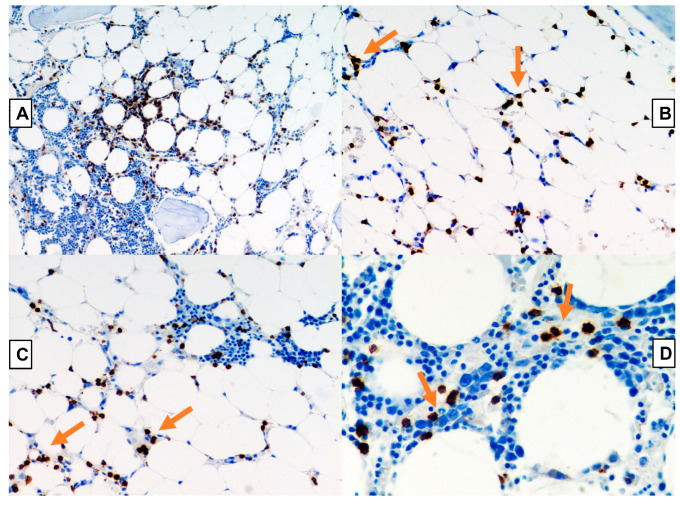
Immunoreactivity for granzyme B; original magnification 100× (**A**) and 200× (**B**). Immunoreactivity for CD3 shows intrasinusoidal bone marrow infiltration; see arrows; respectively original magnification 200× (**C**) and 400× (**D**).

**Figure 5 ijms-23-13392-f005:**
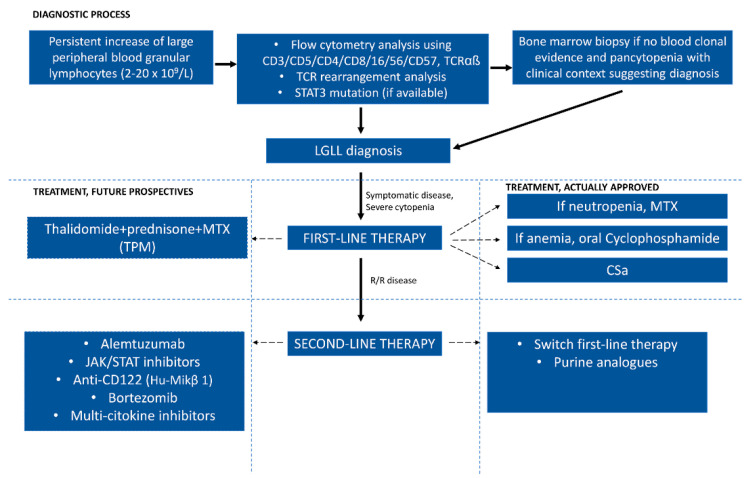
Diagnostic criteria and treatment options in LGL leukemia.

**Table 2 ijms-23-13392-t002:** Autoimmune conditions associated with LGL leukemia.

Disease	Prevalence	Reference
Rheumatoid arthritis	11–36%	[1,35,36]
Vasculitis	2–3%	[42]
Chronic inflammatory bowel disease	2–4%	[37]
Gougerot–Sjögren’s syndrome	2%	[1]
Felty’s syndrome	5%	[1,35,36]
Rhizomelic pseudopolyarthritis	Rare	[7]
Poly/multineuritis	2–3%	[37]
Endocrinopathy	2–14	[35,36]
Precapillary pulmonary hypertension	<0.5%	[43]
Celiac disease	Rare	[29]

**Table 3 ijms-23-13392-t003:** Hematological disease associated with LGL leukemia.

Disease	Prevalence	Reference
Myelodysplastic syndromes	9%	[19,38,51]
Aplastic anemia	4%	[19,44,48,49]
Pure red cell aplasia	7%	[44,47]
Paroxysmal nocturnal hemoglobinuria	Rare	[49]
Myeloproliferative neoplasm	Rare	[33,50]
B-cell neoplasm	Rare	[40]

## Data Availability

The datasets generated during the current study are available from the corresponding author upon reasonable request.

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
