# Peer review of "Molecular Features and Diagnostic Challenges in Alpha/Beta T-Cell Large Granular Lymphocyte Leukemia"

_ijms, 2022, doi:10.3390/ijms232113392_

Round 1
Reviewer 1 Report
This paper from Francesco Gaudio et al provides a clinical picture of two cases probably suffering from T-Large Granular Lymphocytic Leukemia (T-LGL). These cases show uncommon and overlooked features of this disease and help to discuss pathogenic, diagnostic and therapeutic aspects. The manuscript is of merit and interest but several aspects deserve further clarification. The most important aspect is the accuracy of the diagnosis in the two clinical vingettes.
- First, in both cases, peripheral blood findings are not shown (specially the indication of increased circulating clonal cytotoxic T cells). As T-LGL diagnosis is often blood based, this must be expanded. Also, follow up data must be included, as T-LGL diagnosis can be evolving. Moreover, bone marrow pathology report and immunohistochemistry along with more images must be provided.
- Second. The first case shows a clonal cytotoxic T cell expansion in bone marrow the context of liver transplantation. While clonal cytotoxic T cells are shown in transplant settings, these cases may not represent bona fide T-LGL. The authors must complete the clinical picture, specially the signs and symptoms leading to the diagnosis, as fever alone is not a typical manifestation of T-LGL. T-LGL cases have been documented following solid organ trasplantation, including liver transplant. See T-cell large granular lymphocyte leukemia in solid organ transplant recipients: case series and review of the literature - PubMed (nih.gov) for example, but reactive clonal expansions are documented.
Third, in the second case, the authors must clarify the diagnosis, as this case may better fit in aplastic anemia (AA) with a dominant T cell clone rather than T-LGL. Clonal T cell expansions are a frequent finding in AA, and predict for a better response to immunosupressive treatment. The authors must complete the clinical information for this case, including the follow up.
Forth, there are a series of manuscripts recapitulating TLGL pathogenesis in a related journal from this group. Cancers | Special Issue : Large Granular Lymphocytic Leukemia: Genomics and Immunome (mdpi.com) and other journals which recapitulate clinical fingdings of T-LGL. Importantly, NK-LGL and T-LGL and even T-LGL of gammadelta origin have different clinical pictures and different molecular associations. The authors should review their manuscript according to this, as clinical and pathogenic differences do occur according to the celular origin of the disease. Doing this, the title can be kept, otherwise, it should be reformulated to (alpha betha) T-LGL.
Fifth, the therapeutic section is rather short, does not fit in the flow of the review and could be out of the scope of the paper (according to the current title). Maybe it should be reformulated in the introduction and pathogenesis sections.
Last, as new classification systems have emerged (both WHO and International Consensus Classification), the authors should include them in the manuscript. Moreover, the paper would benefit from discussing them along with former diagnostic criteria and how the cases fit in the classification systems.
Author Response
|
First, in both cases, peripheral blood findings are not shown (specially the indication of increased circulating clonal cytotoxic T cells). As T-LGL diagnosis is often blood based, this must be expanded. Also, follow up data must be included, as T-LGL diagnosis can be evolving. Moreover, bone marrow pathology report and immunohistochemistry along with more images must be provided. |
Results from peripheral blood (in particular circulating cytotoxic T cells) have been entered in both cases and follow-up data added; new bone marrow images were inserted |
|
Second. The first case shows a clonal cytotoxic T cell expansion in bone marrow the context of liver transplantation. While clonal cytotoxic T cells are shown in transplant settings, these cases may not represent bona fide T-LGL. The authors must complete the clinical picture, specially the signs and symptoms leading to the diagnosis, as fever alone is not a typical manifestation of T-LGL. T-LGL cases have been documented following solid organ trasplantation, including liver transplant. See T-cell large granular lymphocyte leukemia in solid organ transplant recipients: case series and review of the literature - PubMed (nih.gov) for example, but reactive clonal expansions are documented. |
A new paragraph has been added on the problem of the development of T-LGL clones in the transplant patient.. |
|
Third, in the second case, the authors must clarify the diagnosis, as this case may better fit in aplastic anemia (AA) with a dominant T cell clone rather than T-LGL. Clonal T cell expansions are a frequent finding in AA, and predict for a better response to immunosupressive treatment. The authors must complete the clinical information for this case, including the follow up. |
The correlation between other hematological diseases and LGL clones is better explained in a new paragraph and the diagnostic difficulty of the second case highlighted with the addition of follow-up data. |
|
Forth, there are a series of manuscripts recapitulating TLGL pathogenesis in a related journal from this group. Cancers | Special Issue : Large Granular Lymphocytic Leukemia: Genomics and Immunome (mdpi.com) and other journals which recapitulate clinical fingdings of T-LGL. Importantly, NK-LGL and T-LGL and even T-LGL of gammadelta origin have different clinical pictures and different molecular associations. The authors should review their manuscript according to this, as clinical and pathogenic differences do occur according to the celular origin of the disease. Doing this, the title can be kept, otherwise, it should be reformulated to (alpha betha) T-LGL. |
The title has been reformulated as alpha beta T-LGL |
|
Fifth, the therapeutic section is rather short, does not fit in the flow of the review and could be out of the scope of the paper (according to the current title). Maybe it should be reformulated in the introduction and pathogenesis sections. |
the therapy section has been expanded and better integrated into the flow of the review |
|
Last, as new classification systems have emerged (both WHO and International Consensus Classification), the authors should include them in the manuscript. Moreover, the paper would benefit from discussing them along with former diagnostic criteria and how the cases fit in the classification systems. |
the new classification system has been incorporated into the manuscript |
Reviewer 2 Report
In this case report/review of Francesco Gaudio et al., the author is trying to describe some essential molecular and physiological aspects of chronic T cell-derived LGL. They believe that discussion will help better understand and manage the disease and hopefully pave the way for a targeted clinical approach.
The present article overall provides a convincing discussion on the topic of LGL leukemia. I hope some suggestions from my site will make the review more practical and relevant for the journal readership.
Suggestions:
The abstract should be reconstructed by adding correctly according to the WHO classification of two/three Subtypes of LGL. Also, mention the clinical outcome of these subtypes. Followed by T cell-derived one and noted why the author chose to review the specific one. Also, at least one or two crucial pathways are involved in this disease.
The author can omit T cytotoxic lymphocytes and include T cell-derived LGL in keywords.
The first paragraph of the introduction needs references to mention.
If possible, discuss cytological and serological features and clonal selection (as the author mentioned in the introduction) in different paragraphs.
All the tables the author provided in disorder-related LGL need an extra column with proper mention of the corresponding references.
It would be nice if the author could provide a table with a short discussion on the correlation of other hematological malignancies associated with LGL.
A pictorial presentation of an algorithm of the diagnostic criteria used for treating T LGL leukemia is highly appreciable.
A map of current (Italian/European settings) therapeutic options depending on diagnostic stages will help the reader better understand the future challenges of tackling this indolent disease.
Can the author provide some discussion about except combined chemotherapy is there any clinical trial at present going on with modern immunotherapy or targeted therapy like Stat inhibitor OPB-31121 or NF-kB inhibitor bortezomib?
Author Response
|
The abstract should be reconstructed by adding correctly according to the WHO classification of two/three Subtypes of LGL. Also, mention the clinical outcome of these subtypes. Followed by T cell-derived one and noted why the author chose to review the specific one. Also, at least one or two crucial pathways are involved in this disease. |
The abstract has been reconstructed according to the indications |
|
The author can omit T cytotoxic lymphocytes and include T cell-derived LGL in keywords. |
the keywords have been changed as indicated |
|
The first paragraph of the introduction needs references to mention. |
the reference to the first paragraph of the introduction has been added |
|
If possible, discuss cytological and serological features and clonal selection (as the author mentioned in the introduction) in different paragraphs. |
the cytological and clonal features have been inserted in different paragraphs |
|
All the tables the author provided in disorder-related LGL need an extra column with proper mention of the corresponding references. |
the bibliographic references of the data entered in the table have been inserted |
|
It would be nice if the author could provide a table with a short discussion on the correlation of other hematological malignancies associated with LGL. |
a table with a short discussion to highlight the correlations between malignant hematological diseases and LGL has been added to the manuscript
|
|
A pictorial presentation of an algorithm of the diagnostic criteria used for treating T LGL leukemia is highly appreciable. |
A pictorial presentation of an algorithm of the diagnostic criteria used for the treatment of T LGL leukemia is included |
|
A map of current (Italian/European settings) therapeutic options depending on diagnostic stages will help the reader better understand the future challenges of tackling this indolent disease. |
a map with treatment options has been added
|
|
Can the author provide some discussion about except combined chemotherapy is there any clinical trial at present going on with modern immunotherapy or targeted therapy like Stat inhibitor OPB-31121 or NF-kB inhibitor bortezomib? |
the therapy paragraph has been extended with the recommended discussion |
Round 2
Reviewer 1 Report
This revised edition has several typographic mistakes and the authors have partially answered the questions, however, clarification of the clinical pictures is still missing. I would suggest to remove them and transform the paper in a pure review paper.
In the new paragraph page 2, line 83: T-LGL leukemias have a constitutive mature post-thymic phenotype. The most frequent 83 phenotype is of constitutively activated T cells (CD31, TCR ab1, CD42, CD5dim, CD81, 84 CD161, CD272, CD282, CD45R02, CD45RA1 and CD571), that are aggressive and gener- 85 ally associated with Stat5b mutations [3,16]. 86 CD41 positive LGL leukemia with or without CD8 coexpression almost never features 87 cytopenias, splenomegaly or autoimmune phenomena and has been associated with 88 Stat5b mutations. T-LGL leukemics show a terminal effector memory phenotype with 89 CD45RA expression and CD62L deficiency. The authors must correct CD numbers as is seems they were copied from elsewhere with references.
Page 5 line 184: Lymphocytosis threshold should be indicated for the authors' laboratory, but 3x10e9/L is not indicative of lymphocytosis in most laboratories.
Page 5, line 212. Lymphocytosis threshold should be indicated for the authors' laboratory, but 3x10e9/L is not indicative of lymphocytosis in most laboratories.
Page 5, line 212, there is a typo.
Author Response
|
This revised edition has several typographic mistakes and the authors have partially answered the questions, however, clarification of the clinical pictures is still missing I would suggest to remove them and transform the paper in a pure review paper. |
The two clinical cases were removed from the manuscript and it was transformed into a pure review. |
|
(CD31, TCR ab1, CD42, CD5dim, CD81, CD161, CD272, CD282, CD45R02, CD45RA1 and CD571): The authors must correct CD numbers as is seems they were copied from elsewhere with references. |
The typographical error has been corrected (when converting it into the template the + and - have been transformed into 1 and 2) |
|
Page 5 line 184: Lymphocytosis threshold should be indicated for the authors' laboratory, but 3x10e9/L is not indicative of lymphocytosis in most laboratories. Page 5, line 212. Lymphocytosis threshold should be indicated for the authors' laboratory, but 3x10e9/L is not indicative of lymphocytosis in most laboratories. Page 5, line 212, there is a typo. |
Clinical cases were eliminated |
Round 3
Reviewer 1 Report
The authors have succesfully answered the queries.
Author Response
Thank you for your input.
Best regards